*In memoriam Prof. Xiao'ou Tang*

# LaneSegNet: Map Learning with Lane Segment Perception for Autonomous Driving

**Tianyu Li**[1,2*] **Peijin Jia**[3*] **Bangjun Wang**[4,2*] **Li Chen**[2] **Kun Jiang**[3] **Junchi Yan**[4] **Hongyang Li**[2]

[1]Fudan University  [2]OpenDriveLab  [3]Tsinghua University  [4]Shanghai Jiao Tong University

## Abstract

A map, as crucial information for downstream applications of an autonomous driving system, is usually represented in lanelines or centerlines. However, existing literature on map learning primarily focuses on either detecting geometry-based lanelines or perceiving topology relationships of centerlines. Both of these methods ignore the intrinsic relationship of lanelines and centerlines, that lanelines bind centerlines. While simply predicting both types of lane in one model is mutually excluded in learning objective, we advocate *lane segment* as a new representation that seamlessly incorporates both geometry and topology information. Thus, we introduce **LaneSegNet**, the first end-to-end mapping network generating lane segments to obtain a complete representation of the road structure. Our algorithm features two key modifications. One is a lane attention module to capture pivotal region details within the long-range feature space. Another is an identical initialization strategy for reference points, which enhances the learning of positional priors for lane attention. On the OpenLane-V2 dataset, LaneSegNet outperforms previous counterparts by a substantial gain across three tasks, *i.e.*, map element detection ($+4.8$ mAP), centerline perception ($+6.9$ DET$_l$), and the newly defined one, lane segment perception ($+5.6$ mAP). Furthermore, it obtains a real-time inference speed of $14.7$ FPS. Code is accessible at https://github.com/OpenDriveLab/LaneSegNet.

## 1 Introduction

Perceiving road information holds significant importance for autonomous vehicles. High-definition (HD) maps serve as a rich resource, providing comprehensive geometry and topology information about the road. However, the widespread deployment of HD maps is often hindered by high annotation costs and limited maintainability. To address these challenges, learning-based online mapping methods (Li et al., 2022a; Liao et al., 2023b; Li et al., 2023c) have been proposed. These approaches allow autonomous vehicles to construct HD maps in real time with onboard sensors and the task of reconstructing the road structure boils down to either map element detection or centerline perception.

The existing literature (Li et al., 2022a; Liao et al., 2023b) focusing on the task of map element detection primarily emphasizes the detection of road geometry, as depicted in Fig. 1(b). The map representation is rather oversimplified. Such a formulation of HD maps results in the omission of critical details, such as lane direction, topological structure, laneline type, *etc.*, which are of vital importance for downstream applications such as trajectory planning. To cope with this caveat, certain post-processing techniques are introduced to restore the missing detailed information. Nevertheless, it could incur additional computation costs, which are particularly challenging given the tight budget for real-time deployment in massive production.

Another line of studies (Can et al., 2021; Li et al., 2023c) concentrate on the centerline perception task, aiming to recognize lane topology, as described in Fig. 1(c). These methodologies formulate centerlines as an abstract or virtual representation of lanes and infer centerline connectivity to construct the lane graph. The resultant graph feeds the subsequent planning module with topological

---

*Equal contribution. Primary contact to Tianyu Li via tianyuli23@m.fudan.edu.cn

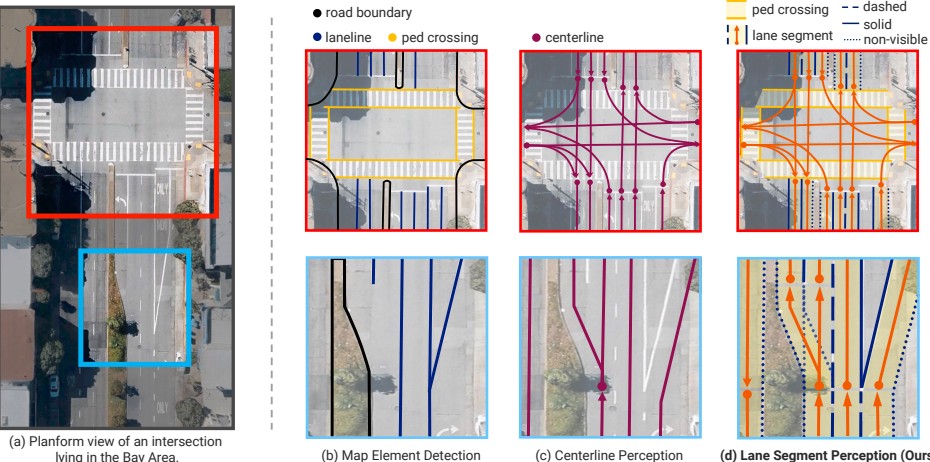

Figure 1: **Comparison of three formulations for online mapping**. **(a)** A realistic scenario of an intersection in the Bay Area, CA. **(b)** Map Element Detection. It detects lanelines, pedestrian crossings, and road boundaries, but fails to adequately indicate topology relationships at intersections and/or situations where lanes diverge/merge. **(c)** Centerline Perception. It depicts lane topology; yet it is a virtual representation and lacks crucial physical geometry information necessary for ensuring safe driving practices. **(d)** Lane Segment Perception. The proposed formulation captures geometric boundaries and semantic lane types in (b), as well as topological graphs in (c). Note that the pedestrian crossing is deemed as a special case (a transverse "lane") in the context of lane segment.

information (Liang et al., 2020). However, it lacks detailed geometry information, which is essential for ensuring safety and precise maneuvering during sophisticated driving behaviors.

We position this work to propose a unified framework that encompasses a comprehensive description of the map information by simultaneously addressing element detection and centerline perception. One straightforward solution is to use two separate branches, one for learning road geometry and the other for modeling lane topology. However, it is empirically observed that such a multi-branch approach turns out to be redundant in handling correlated tasks.

To this end, we propose a new mapping formulation called *lane segment perception* as an alternative. As illustrated in Fig. 1(d), lane segment includes the geometric boundaries, and retains the directed topology connection required to construct a lane graph. Meanwhile, it inherently carries semantic information by encoding line types (dashed, solid, non-visible). Such line types could indicate the crossability of a lane boundary. By doing so, we are able to enhance the complete description of online mapping via interactions with a wide diversity of road information.

To embrace the lane segment philosophy, this work introduces an online end-to-end mapping network, namely **LaneSegNet**, to leverage both geometry and topology modeling. LaneSegNet consists of three parts, encoder, decoder, and predictor. These components work together to predict lane segments and reconstruct the complete road structure. Due to the extra demand to recognize lane segments in challenging scenarios, we devise two non-trivial modifications in the decoder. The first is a lane attention module to capture important local details within the long-range feature space. This is achieved through a heads-to-regions mechanism. The second is an identical initialization strategy for reference points. In contrast to the conventional practice of distributed initialization, where each query is initialized with multiple reference points, such a maneuver excels at learning positional priors for lane attention.

LaneSegNet is validated to be effective on three tasks, *i.e.*, map element detection, centerline perception, and the newly proposed lane segment perception on OpenLane-V2 dataset (Wang et al., 2023). A unanimous improvement of $4.8$ mAP, $9.6$ OLS, and $5.6$ mAP for three tasks has been witnessed respectively. Furthermore, LaneSegNet exhibits superior efficiency for online inference (14.7 FPS) with decoder latency decreased by $31.4\%$ compared to the multi-branch baseline. It is deemed to have the potential to further decrease post-processing time for massive production in autonomous driving. The primary contributions are summarized as follows:

- We introduce lane segment perception as a new map learning formulation. It incorporates both geometry and topology essentials. We hope it would bring valuable insights to the community.
- We propose LaneSegNet, an end-to-end network curated for lane segment perception. Two novel modifications have been proposed, including a lane attention module with a heads-to-regions mechanism for capturing long-range attention, and an identical initialization strategy for reference points to enhance the learning of positional priors for lane attention.

## 2 RELATED WORK

**Centerline Perception:** Centerline Perception (identical to lane graph learning in this paper) from vehicle-mounted sensor data has garnered significant attention recently. STSU (Can et al., 2021) proposes a DETR-like network to detect centerlines, followed by a multi-layer perceptron (MLP) module to determine their connectivity. Building upon STSU, Can et al. (2022) introduce additional minimal cycle queries to ensure the proper order of overlapping lines. CenterLineDet (Xu et al., 2023) treats centerlines as vertices and designs a graph-updating model trained through imitation learning. Notably, Tesla (2022) proposes the concept of "language of lanes" to represent the lane graph as a sentence. Their attention-based model recursively predicts lane tokens and their connectivity. In addition to these piece-wise methods, LaneGAP (Liao et al., 2023a) introduces a path-wise approach to recover the lane graph using an additional conversion algorithm. Aiming at a complete and diverse driving scene graph, TopoNet (Li et al., 2023c) explicitly models the connectivity of centerlines within the network and incorporates traffic elements into the task. In this work, we adopt the piece-wise methodology to construct lane graphs. However, we distinguish ourselves from previous approaches in modeling lane segments instead of centerlines as the vertices of the lane graph, which allows for convenient integration of segment-level geometry and semantic information.

**Map Element Detection:** In prior works (Chen et al., 2022; Huang et al., 2023), attention was paid to elevating map element detection from the camera plane to the 3D space to overcome the projection error. With the trending popularity of BEV perception (Li et al., 2023b), recent works (Philion & Fidler, 2020; Li et al., 2022b; Zhou & Krähenbühl, 2022; Hu et al., 2023; Gao et al., 2023; Liao et al., 2023c) focus on learning HD maps with segmentation and vectorized methods. Map segmentation predicts the semantic meaning of each BEV grid, such as lanelines, pedestrian crossings, and drivable areas. These works differentiate from each other mainly in the perspective view (PV) to BEV transform module. However, the segmented map cannot provide direct information to be employed by downstream modules. HDMapNet (Li et al., 2022a) handles the problem by grouping and vectorizing the segmented map with complicated post-processings.

Though dense segmentation provides pixel-level information, it still cannot touch down the complex relationship of overlapping elements. VectorMapNet (Liu et al., 2023) proposes directly representing each map element as a sequence of points, using coarse key points to decode laneline locations sequentially. MapTR (Liao et al., 2023b) explores a unified permutation-based modeling approach for the sequence of points to eliminate the modeling ambiguity and improve performance and efficiency. PivotNet (Ding et al., 2023) further models map elements with pivot-based representation in a set prediction framework to reduce redundancy and improve accuracy. StreamMapNet (Yuan et al., 2024) employs multi-point attention and temporal information to facilitate the stability of long-range map element detection. In fact, since vectorization also enriches the direction information for lanelines, vectorization-based methods could be easily adapted to centerline perception by alternating the supervision. In this work, we propose a unified and learning-friendly representation, lane segment, for all HD map elements on the road.

## 3 METHODOLOGY

### 3.1 TASK STATEMENT OF LANE SEGMENT PERCEPTION

An instance of the lane segment incorporates both geometric and semantic aspects of the road. As for geometry, it can be represented as a line section composed of the vectorized centerline and its corresponding lane boundaries: $\mathcal{V} = \{\mathcal{V}_{\text{center}}, \mathcal{V}_{\text{left}}, \mathcal{V}_{\text{right}}\}$. Each line is defined as an ordered set of $N_p$ points in 3D space. Moreover, the geometry can also be depicted as a closed polygon that defines the drivable area within this lane.

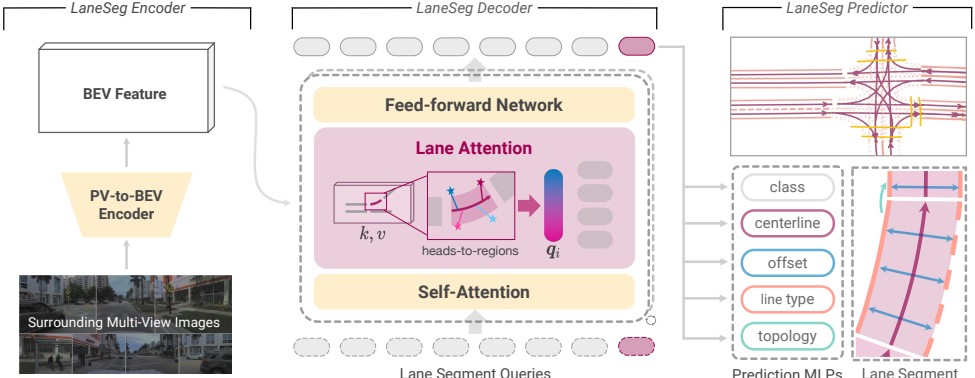

Figure 2: **Pipeline of LaneSegNet**. LaneSegNet is the first end-to-end network curated for the lane segment perception task. It consists of three parts. The LaneSeg encoder generates bird's-eye-view (BEV) features. The core LaneSeg decoder adopts our proposed lane attention module, which employs a heads-to-regions mechanism as well as an identical initialization strategy for reference points. The LaneSeg predictor assembles the elements predicted from different MLPs to form a series of lane segments, thus generating a complete description of the road structure.

In terms of semantics, it includes lane segment classification $\mathcal{C}$ (*e.g.*, lane segment, pedestrian crossing) and line type of the left/right lane boundary (*e.g.*, non-visible, solid, dashed): $\{\mathcal{A}_{\text{left}}, \mathcal{A}_{\text{right}}\}$. These details provide autonomous vehicles with essential insights regarding deceleration requirements and the feasibility of lane changes.

Additionally, topological information plays a vital role in path planning. To represent this information, a lane graph denoted as $\mathcal{G} = (V, E)$ is constructed for lane segments. Each lane segment serves as a node in this graph, represented by the set $V$, while the edges in set $E$ depict the connectivity among lane segments. We use an adjacency matrix to store this lane graph, where the matrix element $(i, j)$ is set to 1 only when the $j$-th lane segment follows the $i$-th one; otherwise, it remains 0.

## 3.2 FRAMEWORK OF LANESEGNET

The overall pipeline of LaneSegNet is depicted in Fig. 2. LaneSegNet takes surrounding view images as input to perceive lane segments in a certain BEV range. In this section, we first brief the LaneSeg encoder for generating the BEV feature. Then, we introduce the lane segment decoder and rmed with lane attention. Lastly, we present the lane segment predictor. and training loss.

### 3.2.1 LANESEG ENCODER

The encoder transforms multi-view images into a BEV feature, denoted as $\mathcal{B} \in \mathbb{R}^{H \times W \times C}$, which is used for lane segment extraction. We utilize a standard ResNet-50 backbone (He et al., 2016) to derive multi-view feature maps from the raw images. Subsequently, the PV-to-BEV encoder module from BEVFormer (Li et al., 2022b) is employed for view transformation.

### 3.2.2 LANESEG DECODER

Transformer-based detection methods utilize decoder to collect features from the BEV feature and update the decoder queries through multiple layers. Each decoder layer utilizes self-attention, cross-attention mechanisms, and a feed-forward network to update queries. Additionally, a learnable positional query is employed. The updated queries are then output and fed to the subsequent stage.

Collecting long-range BEV features is crucial for the online mapping task, owing to complex and elongated map geometry. Prior works (Liao et al., 2023b; Luo et al., 2023) utilize hierarchical (instance-point) decoder queries and deformable attention (Zhu et al., 2020) to extract the local feature for each point query. While this method evades capturing long-range information, high computational cost ensues due to the increased number of queries.

As an instance representation of lanes to construct the scene graph, lane segment obsesses a superior characteristic on instance-level. Instead of using multiple point queries, we aim at adopting a single instance query to represent the lane segment. Consequently, the core challenge lies in how to use a single instance query to cross-attend to the global BEV feature.

**Lane Attention:**  In object detection, deformable attention leverages the positional prior of object and attends only to a small set of attention values near object's *reference point* as a pre-filter, greatly accelerating convergence (Zhu et al., 2020). The reference point is placed at the centriod of predicted object during layer iteration to refine the *sampling locations* of attention values, which are dispersed around the reference point via learnable sampling offset. The intentional initialization of sampling offsets incorporates geometry prior of 2D objects. By doing so, the features from each direction are captured well with the multi-branch mechanism, as shown in Fig. 3(a).

In the context of map learning, Li et al. (2023c) use naive deformable attention to predict centerline. However, as depicted in Fig. 3(b), it may fall short at long-range attention with the naive placement of reference points. Furthermore, due to our target's elongated shape and complex visual cues (*e.g.*, precisely predict the breaking point between solid and dashed laneline), this process requires additional adaptive design for our task. Taking all of those characteristics into consideration, it is necessary for networks to possess the ability to not only attend to long-range contexts but also accurately extract local details. Accordingly, it is advisable to distribute sampling locations across a large region to effectively perceive long-range information. On the other hand, local de-

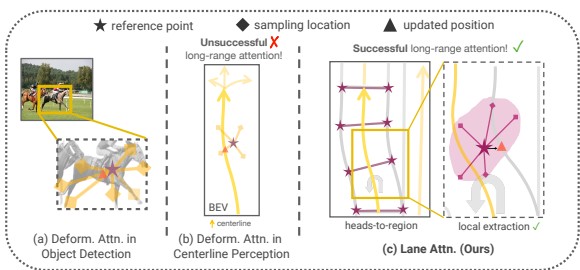

Figure 3: **Advantage of lane attention**. **(a,b)** Each head with native reference point extracts features from various shape-irrelevant directions, which accounts for why deformable attention can not perform long-range attention for line-shape objects. **(c)** By the aid of the heads-to-regions mechanism, our lane attention module effectively gathers long-range context along the elongated lane shape and preserve local details.

tails should remain easily distinguishable for identifying the pivot points. Notably, while competitive interactions are observed among value features within an attention head, the value features between different heads still remain distinct. Therefore, it is promising to explicitly utilize this attribute to facilitate the attention to local features in specific regions.

To this end, we propose a *heads-to-regions* mechanism. We begin by distributing multiple reference points uniformly within the region of lane segment. The sampling locations are then initialized revolving around each reference point in a local region. To preserve the intricate local details, we utilize the multi-branch mechanism, where each head attends to a specific set of sampling locations within a local region, as illustrated in Fig. 3(c).

Taking into consideration the intuitions above, we now provide a mathematical description of the lane attention module. Given BEV feature $\mathcal{B} \in \mathbb{R}^{H \times W \times C}$, the $i$-th lane segment query feature $\boldsymbol{q}_i \in \mathbb{R}^C$ and a set of reference points $\boldsymbol{p}_i \in \mathbb{R}^{M \times 2}$ as input, the lane attention is calculated by:

$$\text{LaneAttn}\left(\boldsymbol{q}_i, \boldsymbol{p}_i, \mathcal{B}\right) = \sum_{m=1}^{M} \boldsymbol{W}_m \left[ \sum_{k=1}^{K} a_{i,m,k} \cdot \boldsymbol{W}'_m \, \text{Bi-linear}\left(\mathcal{B}, \boldsymbol{p}_{i,m} + \Delta \boldsymbol{p}_{i,m,k}\right) \right], \quad (1)$$

where $m$ indexes the attention head, $k$ indexes the sampling locations. The $\boldsymbol{W}'_m \in \mathbb{R}^{C_v \times C}$ and $\boldsymbol{W}_m \in \mathbb{R}^{C \times C_v}$ are learnable weights that project the value feature to a split channel ($C_v = C/M$) and combine the multi-branch feature, thus keep the information of each head. The attention weights $a_{i,m,k}$ is predicted via a linear projection of $\boldsymbol{q}_i$, and normalized as $\sum_{k=1}^{K} a_{i,m,k} = 1$ with soft-max function. The final sampling location is combined of reference point $\boldsymbol{p}_{i,m}$ and sampling offset $\Delta \boldsymbol{p}_{i,m,k}$, which is also predicted via a linear projection of $\boldsymbol{q}_i$. The sampling offset is intentionally initialized to ensure that $K$ sampling locations are evenly centered around the reference points as prior. The flexibility of learnable sampling offset is still reserved, allowing it to attend to details in the local region or capture information from long-range contexts. More implementation details are given in Appendix A.1.3.

**Identical Initialization of Reference Points:** The placement of reference points is determinant to the functionality of lane attention module. In order to align each instance query's attention area with its actual geometry and position, the reference points $p$ of each instance query are distributed based on the lane segment prediction from the previous layer, as illustrated in Fig. 3(c), and the prediction is iteratively refined as well, inspired by Zhu et al. (2020).

Previous work hold that the reference points provided to the first layer ought to be separately initialized with the learnable prior deduced from the positional query embedding. However, since positional query is unrelated to the input images, such initialization method could conversely confine the model's capability of memorizing both the geometrical and positional priors and the falsely generated initialization positions will also pose an obstacle for training.

As such, for the first layer of the lane segment decoder, we propose an identical initialization strategy. In the first layer, every head adopts an identical reference point generated from the positional query. Compared to the distributed initialization of reference points as conventional approaches do, *i.e.*, initializing multiple reference points for each query, identical initialization would make the learning of positional prior more stable by filtering out the distraction of intricate geometries. To be noticed, the identical initialization is seemingly counter-intuitive and yet observed to be effective. As detailed in Sec. 4.3, a performance drop has been witnessed in the ablations about using the distributed initialization.

### 3.2.3 LANESEG PREDICTOR

We employ MLPs in multiple prediction branches to generate the final predicted lane segment from the lane segment query, taking into account geometric, semantic, and topological aspects.

For geometry, we first devise a centerline regression branch to regress the vectorized points position of centerline in 3D coordinates. The output $\mathcal{V}_{\text{center}} \in \mathbb{R}^{N_p \times 3}$ is formatted as $[(x_0, y_0, z_0), ...]$. Owing to the symmetry of the left and right lane boundaries, we introduce an offset branch to predict the offset $\mathcal{V}_{\text{offset}} \in \mathbb{R}^{N_p \times 3}$, which is formatted as $[(\Delta x_0, \Delta y_0, \Delta z_0), ...]$. Thus, the coordinates of left and right lane boundaries can be calculated with $\mathcal{V}_{\text{left}} = \mathcal{V}_{\text{center}} + \mathcal{V}_{\text{offset}}$ and $\mathcal{V}_{\text{right}} = \mathcal{V}_{\text{center}} - \mathcal{V}_{\text{offset}}$.

Given that the lane segment can be conceptualized as the drivable area, we integrate an instance segmentation branch into the predictor. The instance mask prediction part follows Cheng et al. (2021). As for the semantics, three classification branches predict the classification score for $\mathcal{C}$, the scores for $\mathcal{A}_{\text{left}}$ and $\mathcal{A}_{\text{right}}$ in parallel. The topology branch takes the updated query features as input and outputs a weighted adjacent matrix for the lane graph $\mathcal{G}$ using MLPs, following Li et al. (2023c).

### 3.3 TRAINING LOSS

Adopting the DETR-like paradigm, LaneSegNet efficiently computes the one-to-one optimal assignment between predictions and ground truth with the Hungarian algorithm. Training losses are then computed based on the assignment results. The loss function is composed of four parts: geometric loss, classification loss, laneline type classification loss, and topological loss.

Geometric loss supervises the geometry of each predicted lane segment. According to the bipartite matching result, each predicted vectorized lane segment $\mathcal{V}$ is assigned with a GT lane segment $\widetilde{\mathcal{V}}$. The vectorized geometric loss is defined as the Manhattan distance computed between the assigned lane segment pairs. Specifically, the vectorized geometric loss is calculated between $\mathcal{V} = \{\mathcal{V}_{\text{left}}, \mathcal{V}_{\text{center}}, \mathcal{V}_{\text{right}}\}$ and $\widetilde{\mathcal{V}} = \{\widetilde{\mathcal{V}}_{\text{left}}, \widetilde{\mathcal{V}}_{\text{center}}, \widetilde{\mathcal{V}}_{\text{right}}\}$ accordingly. Taking full advantage of geometric constraints, the explicit manner of supervision on left and right lane boundaries can further benefit the regression of centerlines. Besides, a composite loss function consisting of a Cross-Entropy loss and a dice loss (Milletari et al., 2016) is applied for the supervision of predicted masks following Cheng et al. (2022)'s manner. As for the supervision for laneline type prediction, we introduce laneline type classification loss which applies cross-entropy loss on $\{\mathcal{A}_{left}, \mathcal{A}_{right}\}$ and $\{\widetilde{\mathcal{A}}_{left}, \widetilde{\mathcal{A}}_{right}\}$ correspondingly. Furthermore, inspired by Li et al. (2023c), we introduce the topological loss, which designates the focal loss (Lin et al., 2017b) calculated based on the topology relationship information $\mathcal{G}$ and $\widetilde{\mathcal{G}}$ and serves to supervise the relationship among lane segments.

Table 1: **Lane segment perception**. LaneSegNet surpasses other approaches by a large margin. For consistency, we opted not to add a topology head to MapTR series or a line type head to either MapTR series or TopoNet, preventing any modifications that might influence the original design.

| Method | mAP↑ | $AP_{ls}$↑ | $AP_{ped}$↑ | $TOP_{lsls}$↑ | $AE_{type}$↓ | $AE_{dist}$↓ | FPS |
|---|---|---|---|---|---|---|---|
| TopoNet | 23.0 | 23.9 | 22.0 | 1.0 | - | 0.769 | 10.5 |
| MapTR | 27.0 | 25.9 | 28.1 | - | - | 0.695 | 14.5 |
| MapTRv2 | 28.5 | 26.6 | 30.4 | - | - | 0.702 | 13.6 |
| **LaneSegNet-tiny** | 28.5 | 28.2 | 28.7 | 6.8 | 10.6 | 0.710 | **16.2** |
| **LaneSegNet** | **32.6** | **32.3** | **32.9** | **8.1** | **9.2** | **0.673** | 14.7 |

## 4 EXPERIMENTS

### 4.1 DATASET AND METRICS

**Dataset:** We evaluate LaneSegNet on the popular OpenLane-V2 dataset (Wang et al., 2023). This dataset contains 1,000 scenes, with each scene lasting approximately 15 seconds. The training set includes about 27,000 frames, and the validation set includes about 4,800 frames. Labels are further elaborately re-annotated in our proposed lane segment manner, and the map data from the Argoverse 2 dataset is further processed to align with our proposed representation. All lane segments within $[-50m, +50m]$ along the x-axis and $[-25m, +25m]$ along the y-axis are annotated in the 3D space. Each line in $\mathcal{V}$ is represented by an ordered set of 10 points during training and evaluation. Details of our data processing approach can be found in Appendix A.2. The data will be released with code.

**Metrics:** Since the models we are comparing are specifically designed for different tasks, we categorize the evaluation metrics into three parts: map element detection, centerline perception, and lane segment perception, to ensure a fair comparison.

The evaluation metrics for centerline perception use existing benchmarks presented in OpenLane-V2 (Wang et al., 2023). Specifically, the $DET_l$ score serves as the average precision (AP) for assessing centerline perception performance, primarily relying on the Fréchet distance. The $TOP_{ll}$ score measures the performance of topology reasoning. The OLS as a weighted sum is also included[1].

In the map element detection benchmark, we adopt AP for lane divider (or laneline) and pedestrian crossing to evaluate the map construction quality under Chamfer distance thresholds of $\{0.5, 1.0, 1.5\}$ meters. The mean AP is computed as the average of the AP values for each class.

Building upon the established metrics for these two mapping tasks, lane segment perception adopts specifically designed metrics. lane segment distance $\mathcal{D}_{ls}$, corresponding average precision $AP_{ls}$ and $AP_{ped}$, the mean AP computed as the average of $AP_{ls}$ and $AP_{ped}$, topology metric $TOP_{lsls}$ and another two average errors $AE_{type}$ and $AE_{dist}$. All of these metrics are defined in Appendix A.4.

### 4.2 MAIN RESULTS

**Lane Segment Perception:** In Tab. 1, we compare LaneSegNet with several state-of-the-art methods, MapTR (Liao et al., 2023b), MapTRv2 (Liao et al., 2023c) and TopoNet (Li et al., 2023c), on the newly introduced lane segment perception benchmark. We re-train their model with our lane segment label. The implementation details can be found in Appendix A.3. LaneSegNet outperforms other methods up to 9.6% on mAP, and the average distance error is relatively reduced by 12.5%. LaneSegNet-tiny also surpasses previous approaches, with a higher FPS of 16.2.

**Qualitative results:** As shown in Fig. 4, LaneSegNet is capable of handling complex intersections. For additional qualitative results, please refer to Appendix C.

**Map Element Detection:** To perform a more fair comparison with the map element detection methods (Liu et al., 2023; Liao et al., 2023b), we decompose the predicted lane segments of Lane-

---

[1]OLS is short for OpenLane Score which is a comprehensive metric defined in OpenLane-V2 (Wang et al., 2023). DET represents the detection metric of certain attributes which are usually given in subscript and calculated in the way of mAP.

Table 2: **Map element detection**. The performance of LaneSegNet is even superior to approaches that were natively trained on this task. *: Reshaping the prediction from the lane segment detection model.

| Method | mAP↑ | AP$_{div}$↑ | AP$_{ped}$↑ |
|---|---|---|---|
| VectorMapNet | 8.3 | 5.5 | 11.1 |
| MapTR | 22.7 | 16.0 | 29.4 |
| **LaneSegNet***  | **27.5** | **21.8** | **33.2** |

Table 3: **Centerline perception**. LaneSegNet outperforms other methods on both centerline perception and topology reasoning. The OLS is calculated between DET$_l$ and TOP$_{ll}$. *: Extracting centerlines from the lane segment results.

| Method | OLS↑ | DET$_l$↑ | TOP$_{ll}$↑ |
|---|---|---|---|
| STSU | 13.2 | 12.6 | 1.9 |
| TopoNet | 20.0 | 24.9 | 2.3 |
| **LaneSegNet***  | **29.7** | **31.8** | **7.6** |

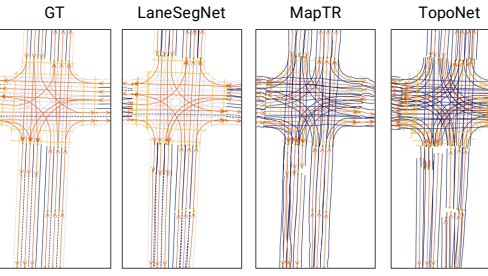

Figure 4: **Qualitative results**. LaneSegNet is capable of reconstructing the topology of intersections and identifying pedestrian crosswalks. Various types of laneline are also categorized well. However, due to limitations in camera perspective, there are errors in estimating the number of lateral lanes on both sides of the intersection.

SegNet into pairs of lanelines and then compare them with state-of-the-art methods using map element detection metrics. We feed the disassembled laneline and pedestrian crossing labels into several state-of-the-art methods for re-training. The experimental results, as presented in Tab. 2, reveal that LaneSegNet consistently surpasses other approaches for the map element detection task. Under a fair comparison, LaneSegNet can restore the road geometry better with extra supervision. This indicates that the lane segment learning representation is adept at capturing road geometric information.

**Centerline Perception:** We also compare LaneSegNet with state-of-the-art centerline perception methods (Can et al., 2021; Li et al., 2023c) in Tab. 3. For consistency, centerlines are also extracted from lane segments for re-training. It could be concluded that LaneSegNet's performance is drastically above other methods in lane graph perception tasks. With additional geographical supervision, LaneSegNet also demonstrates superior topology reasoning ability. It proves that reasoning ability is closely correlated with strong localization and detection ability.

### 4.3 ABLATION STUDY

**Lane Segment Formulations:** In Tab. 4, we provide ablations to verify the designing merits and training efficiency of our proposed lane segment learning formulation. Compared with separated training models on the first two rows, the joint training of both centerline and map element brings an all-rounded and on average 1.3 improvement for the two main metrics as is presented in row 4, proving the feasibility of multi-task training. However, the vanilla approach to training both centerline and map element in a single branch by adding extra categories leads to an obvious performance drop. Our model trained with lane segment label gains salient performance enhancement over vanilla single branch approaches mentioned above ($+7.2$ on OLS and $+4.4$ on mAP for comparison between row 3 and row 5) , which validates the positive interaction among various road information in our map learning formulation. Our model even surpasses the multi-branch approach, particularly in centerline perception ($+4.8$ on OLS). This suggests that geometry can guide topology reasoning in our map learning formulation, where the multi-branch model only marginally excels over the CL-only model ($+0.6$ OLS between row 1 and row 4). As for the minor decrease on AP$_{div}$, it comes from the reshaping process of our prediction results and is caused by the error on line type classification,

Table 4: **Comparison among different learning formulations**. "CL-only" (or "ME-only") refers to using a single branch to learn the centerline perception (or map element detection) task. "CL + ME" denotes a multi-task formulation to perform both tasks. All the models follow the same architecture without our specific designs mentioned in Sec. 3.2.2.

| Formulation | OLS | $DET_l$ | $TOP_{ll}$ | mAP | $AP_{div}$ | $AP_{ped}$ | Latency (ms) | # Param. |
|---|---|---|---|---|---|---|---|---|
| centerline (CL) only | 21.1 | 22.9 | 3.7 | - | - | - | 16.29 | 5.64M |
| map element (ME) only | - | - | - | 21.8 | 17.3 | 26.2 | 14.41 | 5.18M |
| CL + ME, single branch | 19.3 | 22.5 | 2.6 | 20.4 | 17.0 | 23.7 | 16.97 | 5.64M |
| CL + ME, multi-branch | 21.7 | 23.6 | 3.9 | 23.9 | **19.3** | 28.4 | 30.14 | 10.81M |
| **Lane Segment** | **26.5** | **27.2** | **6.6** | **24.8** | 19.1 | **30.5** | 20.67 | 6.74M |

Table 5: **Comparison among different cross-attention designs**. "Deform. Attn." refers to deformable attention, "inst." means using instance query, "hie." refers to hierarchical query design.

Table 6: **Ablation on the designs in LaneSeg decoder**, including heads-to-regions (heads2r.) and identical initialization (ident. init.) of reference points.

| Method | $n_{ref}$ | mAP | $TOP_{lsls}$ |
|---|---|---|---|
| Deform. Attn. (inst.) | 1 | 28.7 | 6.9 |
| Deform. Attn. (hie.) | $N_p$ | 29.1 | 5.2 |
| Multi-Point Attn. | $N_p$ | 24.6 | 6.9 |
| **Lane Attn. (Ours)** | 8 | **32.6** | **8.1** |

| Exp. | heads2r. | ident. init. | mAP | $TOP_{lsls}$ |
|---|---|---|---|---|
| 1 | | | 28.8 | 5.6 |
| 2 | ✓ | | 29.1 | 6.2 |
| 3 | | ✓ | 30.6 | 7.9 |
| 4 | ✓ | ✓ | **32.6** | **8.1** |

which is fully acceptable. Besides, the latency of the model adopting lane segment is 10ms quicker than the multi-branch approach (31.4% latency improvement) and the number of parameters is also reduced by 37.7%, proving that our learning formulation has stricken the balance between robust performance and training efficiency.

**Lane Attention Module:** Ablations of our presented attention module are presented in Tab. 5. In order to facilitate a fair comparison, we substitute the lane attention module in our framework with alternative attention designs. With our elaborated designs, LaneSegNet with lane attention significantly outperforms these approaches, showing a substantial improvement (+3.9 on mAP and +1.2 on $TOP_{ll}$ compared to row 1). Additionally, compared to the hierarchical query design, the decoder latency can be further reduced (from 23.45ms to 20.96ms), due to a reduction in the number of queries.

Specifically, the ablations on several designs proposed in Sec. 3.2.2 are showcased in Tab. 6. The heads-to-regions mechanism consistently enhances the geometry prediction, yielding a +2.0 mAP improvement when comparing row 3 to row 4. This demonstrates its superior ability to preserve local details. Furthermore, identical initialization improves mAP by 3.5 by accelerating the convergence. The full results for ablations and more details are provided in the Appendix B.2.2.

## 5 CONCLUSION

In this paper, we present lane segment perception as a new map learning formulation and propose LaneSegNet, a tailored end-to-end network to address the problem. Along with the network, two innovative enhancements are proposed, including a lane attention module, which employs a heads-to-regions mechanism to capture long-range attention, and an identical initialization strategy for reference points to enhance the learning of positional priors for lane attention. Experiment results on the OpenLane-V2 dataset demonstrate the effectiveness of our designs.

**Limitations and Future Work.** Due to the computational limitation, we have not expanded the proposed LaneSegNet to more additional backbones. More popular datasets mentioned in Li et al. (2023a), such as nuScenes and Waymo are not incorporated, due to the missing lane segment annotation. The formulation of lane segment perception and LaneSegNet can potentially benefit downstream tasks which is worth future exploration.

## ACKNOWLEDGEMENTS

OpenDriveLab is the autonomous driving team affiliated with Shanghai AI Lab. This work was supported by National Key R&D Program of China (2022ZD0160104), NSFC (62206172), and Shanghai Committee of Science and Technology (23YF1462000). We extend our gratitude to Huijie Wang, Chonghao Sima, and Jia Zeng for their profound discussions.

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

## *Appendix*

## A IMPLEMENTATION DETAILS

### A.1 LANESEGNET

#### A.1.1 BEV FEATURE EXTRACTOR

To obtain image features, we employ a ResNet-50 (He et al., 2016) with an FPN (Lin et al., 2017a). Three stages of multi-scale feature maps from ResNet-50 are inputted into FPN for fusion and combination. These three features are the downsampling results of the image at scales $S_{8\times}$, $S_{16\times}$, $S_{32\times}$. The final output features have four levels with an additional $S_{64\times}$ level and each level has 256 output channels. Afterward, we utilize three encoder layers, as proposed in BEVFormer (Li et al., 2022b) to transform the multi-scale image features into BEV features. The final BEV grid is set to $200 \times 100$, corresponding to the perception range of $\pm 50$ m $\times \pm 25$ m.

#### A.1.2 LANE SEGMENT DECODER

In the decoder, we utilize six decoder layers based on Deformable DETR (Zhu et al., 2020), which consists by of three components: a self-attention layer, a cross-attention layer, and a feed-forward network (FFN). The self-attention layer employs 8 attention heads. The cross-attention uses a lane attention module which also has 8 attention heads and incorporates 32 offset points around reference points. We adopt a two-layer FFN with a feed-forward channel size of 512. The initial query embeddings represented as $q = [q_p, q_i]$, consists of 256 channels $q_p$ for generating the initial reference point and the other 256 channels $q_i$ for the initial instance query. The number of queries is set to 200 for lane segments.

#### A.1.3 LANE ATTENTION

In the process of heads-to-regions, it is critical to guarantee that reference points are uniformly dispersed within a lane segment. To achieve this, we methodically allocate four reference points along two predicted boundaries of a lane segment. Consequently, lane attention is directed to each local area along the lane segment through distinct heads, enabling long range attention and precise local feature extraction. During the sampling offset initialization stage, for each reference point, the number of sampling locations K is set to 32. We center around a reference point, and distribute four sampling locations from near to far in each of the eight directions, following Deformable DETR's (Zhu et al., 2020) manner.

#### A.1.4 LANE SEGMENT PREDICTOR

The three classification branches all consist of a three-layer MLP with LayerNorm and ReLU in between, which predicts the confidence score of the lane segment class and its left/right line type. The two regression heads for centerline and offset are both three-layer MLPs with ReLU, which predicts the ordered point set of $10 \times 3$ for the centerline and its corresponding offset. For mask prediction, we use an MLP to convert the query embeddings $q \in \mathbb{R}^{N \times \delta}$ into mask embeddings $\mathcal{E}_{\text{mask}} \in \mathbb{R}^{N \times C}$. Then the $N$ possibly instance mask predictions $S$ can be obtained via a dot product between mask embeddings $\mathcal{E}_{\text{mask}}$ and BEV features $\mathcal{B}$, followed by a sigmoid activation.

$$S = \text{sigmoid}(\mathcal{E}_{\text{mask}} \otimes \mathcal{B}). \tag{2}$$

The topology branch takes the matched query features $Q$ as input and outputs a weighted adjacent matrix $A$ for the lane graph $\mathcal{G}$. For each pair of $q_i, q_j \in Q$, the weight of edge $A_{i,j}$ can be calculated as follows:

$$\begin{aligned} q_i' &= \text{MLP}_{\text{pre}}(q_i), q_j' = \text{MLP}_{\text{suc}}(q_j), \\ A_{i,j} &= \text{sigmoid}\left(\text{MLP}_{\text{top}}\left(\text{concat}(q_i', q_j')\right)\right), \end{aligned} \tag{3}$$

where the parameters are not shared between MLPs for predecessors and for successors, following Li et al. (2023c).

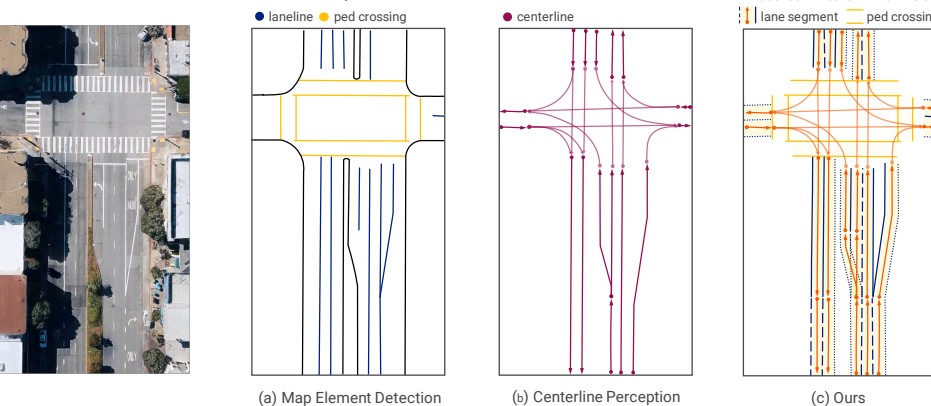

Figure 5: **Different online mapping formulations**.

## A.2 DATA PROCESSING

The conventional data format for map element detection, as illustrated in Fig. 5(a), includes visible road geometry such as road boundary, laneline, and pedestrian crossing. On the other hand, the centerline perception, depicted in Fig. 5(b), consists of centerlines and their corresponding topology relationships. In our lane segment formulation, as shown in Fig. 5(c), we combine these two formats and introduce line types to create a comprehensive representation. The line type is extremely important for a general agent to make driving decisions (Sima et al., 2023).

We construct the lane segment data based on the Argoverse 2 dataset (Wilson et al., 2021). However, the map in Argoverse 2 contains unnecessary breakpoints that do not conform to actual driving rules. To address this problem, we apply a depth-first search (DFS) merging process inspired by OpenLane-V2 (Wang et al., 2023). However, unlike OpenLane-V2, as our algorithm takes into account both topology relationships and line types, the search boundaries for merging expand to encounter diverge, merge, intersection, and changes in the laneline type. Additionally, our data processing applies not only to the centerline but also to the lanelines of the lane instance, treating them as a whole during the merging process. Regarding line types, while the data processing currently focuses on the most fundamental line types, such as dashed, solid, and non-visible, it is also flexible enough to support the addition of other line types if required.

Considering the richness of online maps, we incorporate pedestrian crossing into data and simplify it into lane segment format. We treat pedestrian crossing as two edges along its principal axis. Due to the inconsistency in the directions of pedestrian crossing, which can negatively affect the training, we uniformly reorganize the direction of the principal axis to face the top-left direction in the BEV coordinate system and both lines should be pointing in nominally the same direction. In the aforementioned processing of lane segments, the boundary information is implicitly included, eliminating the need for separate processing of this class.

Furthermore, we migrate our data to two subtasks: map detection and centerline perception. For map element detection, to maintain consistency with previous conventional algorithms, we extract visible lanelines from lane segments for learning. For centerline perception, we extract the corresponding centerline for learning. Additionally, to align with the learning of TopoNet, the new relationships between traffic elements and centerlines are obtained by finding the corresponding relationship between the centerline in OpenLane-V2 and the lane segment, and then modifying the corresponding matrix.

## A.3 TRAINING DETAILS

In the BEV feature extraction module, we initially use ResNet-50 (He et al., 2016) as the foundational image backbone to extract image features. Subsequently, we apply FPN (Lin et al., 2017a) to obtain multi-level feature maps. Following this, we incorporate 3 encoder layers to facilitate view transformation. Within the lane segment detection module, we utilize a methodology based on the

Deformable DETR structure, incorporating a stack of 6 decoder layers. During the training process, we employ the AdamW optimizer (Loshchilov & Hutter, 2018) in conjunction with a cosine annealing schedule. The initial learning rate is set to $2 \times 10^{-4}$. LaneSegNet is trained using 8 NVIDIA Tesla V100 GPUs, with a total batch size of 8, over the course of 24 training epochs. Regarding the loss, the final loss function is defined as:

$$\mathcal{L} = \lambda_{vec}\mathcal{L}_{vec} + \lambda_{seg}\mathcal{L}_{seg} + \lambda_{cls}\mathcal{L}_{cls} + \lambda_{type}\mathcal{L}_{type} + \lambda_{top}\mathcal{L}_{top}, \tag{4}$$

where $\mathcal{L}_{seg} = \lambda_{ce}\mathcal{L}_{ce} + \lambda_{dice}\mathcal{L}_{dice}$, and the hyperparameters are defined as: $\lambda_{vec} = 0.025, \lambda_{seg} = 3.0, \lambda_{ce} = 1.0, \lambda_{dice} = 1.0, \lambda_{cls} = 1.5, \lambda_{type} = 0.01, \lambda_{top} = 5.0$.

To be specific, adopting the DETR-like paradigm, LaneSegNet infers a fixed number of $N$ predictions in a single pass through the decoder, and the bipartite matching acts as the prerequisite for instance-specific loss optimization. We denote by $y$ the ground truth of lane segments, and $\hat{y}$ the $N$ predicted segment sets. A bipartite matching can be formulated as an optimization problem that requires finding a permutation of $N$ elements with the lowest cost:

$$\hat{\sigma} \triangleq \underset{\sigma \in \mathfrak{S}^N}{\arg\min} \sum_i^N \mathcal{L}_{\text{match}}(y_i - \hat{y_{\sigma_i}}), \tag{5}$$

where $\mathfrak{S}^N$ represents the vector space composed of all permutations of $N$ elements and $\mathcal{L}_{\text{match}}(y_i, \hat{y_{\sigma_i}})$ is the pair-wise matching cost between ground truth $y_i$ and a prediction with index $\sigma_i$. This optimized assignment can be efficiently computed with the Hungarian algorithm or Sinkhorn algorithm, following DETR's designing ideology.

In light of so many properties embodied by a single lane segment, one of the primary training difficulties lies in finding a proper definition of matching cost between predictions and ground truths of lane segments. In LaneSegNet, our matching cost takes into account both the semantic classification and the similarity of predicted and ground truth lane segments.

For the semantic classification part, the matching cost is composed of both classification loss $\mathcal{L}_{cls}$ and laneline type classification loss $\mathcal{L}_{type}$. $\mathcal{L}_{cls}$ employs focal loss (Lin et al., 2017b) on the predicted class score $\mathcal{C}$ and the actual class label $\widetilde{\mathcal{C}}$. $\mathcal{L}_{type}$ applies cross-entropy loss on $\{\mathcal{A}_{left}, \mathcal{A}_{right}\}$ and $\{\widetilde{\mathcal{A}}_{left}, \widetilde{\mathcal{A}}_{right}\}$ correspondingly.

$$\mathcal{L}_{cls} = \sum_{i=0}^{N-1} \mathcal{L}_{\text{Focal}}(\widetilde{\mathcal{C}}_{\hat{\sigma}_i}, \mathcal{C}_i). \tag{6}$$

The similarity between two lane segments can be measured from two dimensions: geographical distance and geometric variance. For the former one, we define $\mathcal{L}_{vec}$ as the $\ell_1$ loss computed on the predicted vectorized lane $\mathcal{V}$ and the GT $\widetilde{\mathcal{V}}$. For the latter one, we apply the composite loss consisting of a Cross-Entropy loss and a dice loss (Milletari et al., 2016) on the predicted instance mask and corresponding GT, represented as $\mathcal{L}_{seg} = \lambda_{ce}\mathcal{L}_{ce} + \lambda_{dice}\mathcal{L}_{dice}$.

As for training loss, we augment our matching cost with a topological loss component $\mathcal{L}_{ll}$, which designates a Focal loss calculated based on the topological relationship information $\mathcal{G}$ and $\widetilde{\mathcal{G}}$ and serves to supervise the topological relationships among lane segments.

### A.4 METRICS FOR LANE SEGMENT PERCEPTION

Building upon the established metrics for these two mapping tasks, we present the evaluation metrics for lane segment perception. First, we define the lane segment distance as a weighted sum of the distances between the left/right lane boundaries and centerlines, also considering the lane direction. The resulting lane segment distance, denoted as $\mathcal{D}_{ls}$, between the prediction $\mathcal{V}$ and the ground truth $\widetilde{\mathcal{V}}$ is expressed as follows:

$$\mathcal{D}_{ls}(\mathcal{V}, \widetilde{\mathcal{V}}) = 0.5 \cdot \left[ \texttt{Chamfer}([\mathcal{V}_{\text{left}}, \mathcal{V}_{\text{right}}], [\widetilde{\mathcal{V}}_{\text{left}}, \widetilde{\mathcal{V}}_{\text{right}}]) + \texttt{Fréchet}(\mathcal{V}_{\text{center}}, \widetilde{\mathcal{V}}_{\text{center}}) \right]. \tag{7}$$

Using the aforementioned distance function, we can calculate an average precision, denoted as $\text{AP}_{ls}$, over three matching thresholds given by $\mathbb{T} = \{1.0, 2.0, 3.0\}$ meters. The $\text{AP}_{ped}$ is determined solely

based on the Chamfer distance, aiming to assess the non-directional pedestrian crossing, which aligns with thresholds in map element detection. The mean AP (mAP) is computed as the average of $AP_{ls}$ and $AP_{ped}$. Additionally, a topology metric, $TOP_{lsls}$, is adapted from $TOP_{ll}$ to assess the similarity between the prediction and ground truth lane graphs. It is defined as the averaged vertice mAP between $(V, E)$ and $(\hat{V}', \hat{E}')$ over all vertices:

$$\text{TOP} = \frac{1}{|V|} \sum_{v \in V} \frac{\sum_{\hat{n}' \in \hat{N}'(v)} P(\hat{n}') \mathbb{1}(\hat{n}' \in N(v))}{|N(v)|}, \tag{8}$$

where $N(v)$ denotes the ordered list of neighbors of vertex $v$ ranked by confidence and $P(v)$ is the precision of the $i$-th vertex $v$ in the ordered list. The $TOP_{lsls}$ is for topology among lane segments on graph $(V_{ls}, E_{lsls})$.

## B EXPERIMENTS

### B.1 BASELINES

For all models re-implemented for all the tasks, we uniformly employ ResNet-50 as the foundational image backbone and apply FPN to generate multi-level feature maps. Regarding the view transformation from surrounding multi-views to the bird's-eye-view, the same encoder adopted from BEVFormer (Li et al., 2022b) is applied.

**Lane Segment Perception:** In the lane segment perception task, MapTR (Liao et al., 2023b), MapTRv2 (Liao et al., 2023c) and TopoNet (Li et al., 2023c) are retrained as our baselines. For implementation, different approaches are applied. Specifically, we modify the prediction of 20 points for a single laneline in MapTR series into predicting 10 points for the left and right boundaries respectively. The centerline is obtained by averaging the positions of the left and right lane lines. Furthermore, since lane segments have a unidirectional nature, we abandon the approach that considers different permutations. For the adaptation of TopoNet, we introduce an additional branch to predict the offset and use it to predict the lane segment as the centerline plus or minus the offset. The traffic element branch is kept for TopoNet. The results of FPS are benchmarked on an NVIDIA A100 GPU.

**Map Element Detection:** In the task of detecting map elements, both MapTR and LaneSegNet are trained using their original methods. However, a slight modification needs to be made regarding the modeling approach for pedestrian crossings, which is changed from polygons to polylines. During the evaluation process, LaneSegNet's predictions are additionally post-processed to convert them into the format of map elements.

**Centerline Perception:** We select STSU and TopoNet as our baseline models for the centerline perception task. To ensure a fair comparison, both STSU and TopoNet are trained under the same conditions with the newly extracted centerline labels.

### B.2 ABLATION STUDY

#### B.2.1 LANE SEGMENT FORMULATIONS

This section introduces the specific content of Tab. 4 in experimental details and evaluation details. To fairly illustrate the efficiency of lane segment formulation, we remove all specific designs and only utilize the basic Deformable DETR architecture for comparison. For centerline (CL) only, it predicts the centerline parts in the lane segment. For map element (ME) only, it predicts the divider and pedestrian crossing extracted from the lane segment. In the combined training setting, "single-head" means using a shared decoder followed by a classification head to determine the prediction belonging, while "multi-branch" indicates using two separate decoders and heads to learn CL and ME respectively. As for Lane Segment, we use our proposed formulation to train and further process the predictions into corresponding CL and ME. Concerning the metric $AP_{div}$, we leverage the lane type from the lane segment to filter out non-visible elements and perform further evaluation. As for the latency and the number of parameters, we can see that compared to ME only, CL only adds

Table 7: **Comparison among different cross-attentions**. "Deform. Attn." refers to deformable attention, "inst." means using instance query, "hie." refers to hierarchical query design.

| Method | inst. query | $n_{ref}$ | mAP↑ | $AP_{ls}$↑ | $AP_{ped}$↑ | $TOP_{lsls}$↑ | $AE_{type}$↓ | $AE_{dist}$↓ |
|---|---|---|---|---|---|---|---|---|
| Deform. Attn. (init.) | ✓ | 1 | 28.7 | 27.8 | 29.6 | 6.9 | 10.9 | 0.705 |
| Deform. Attn. (hie.) | | $N_p$ | 29.1 | 29.1 | 29.2 | 5.2 | 8.6 | **0.668** |
| Multi-Point Attn. | ✓ | $N_p$ | 24.6 | 26.1 | 23.2 | 6.9 | 12.9 | 0.711 |
| **Lane Attn. (Ours)** | ✓ | 8 | **32.6** | **32.3** | **32.9** | **8.1** | **9.2** | 0.673 |

Table 8: **Ablation on the designs in LaneSeg decoder**, including heads-to-regions (heads2r.) and identical initialization (ident. init.) of reference points.

| Exp. | *heads2r.* | *ident. init.* | mAP↑ | $AP_{ls}$↑ | $AP_{ped}$↑ | $TOP_{lsls}$↑ | $AE_{type}$↓ | $AE_{dist}$↓ |
|---|---|---|---|---|---|---|---|---|
| 1 | | | 28.8 | 28.7 | 28.9 | 5.6 | 9.7 | 0.692 |
| 2 | ✓ | | 29.1 | 28.6 | 29.6 | 6.2 | 9.4 | 0.687 |
| 3 | | ✓ | 30.6 | 30.6 | 30.7 | 7.9 | 9.4 | 0.677 |
| 4 | ✓ | ✓ | 32.6 | 32.3 | 32.9 | 8.1 | 9.2 | 0.673 |

a topology head which increases the model size. These metrics for the multi-branch model are double compared to the single-task model. Additionally, due to the inclusion of offset and line type predictions in the lane segment, its performance in these two aspects is slightly higher compared to the single-task model.

### B.2.2 Lane Attention Module

Tab. 7 shows the detailed metrics of the ablation on the cross-attention module. Initially, we employ a native instance query design of deformable attention as baseline, with only one reference point placed at the centroid of lane segment. Opting for the hierarchical query design (Liao et al., 2023b) results in a marginal improvement in the mAP ($+0.4$ for comparison between row 1 and row 2), but the topology reasoning performance drops, which can be attributed to either inadequate instance representation or point query issues. Though slightly lagging behind in $AE_{dist}$. This may be attributed to the fact that the hierarchical query provides more learnable space within the internal structure of the lane. In a concurrent work, Yuan et al. (2024) propose Multi-Point Attention, which utilizes predicted geometry to adjust the placement of reference points and employs instance queries. However, their approach merely relies on distributed initialization for reference points without considering the offset of sampling points. This leads to a 4.1 decline in mAP compared to the baseline, suggesting that indiscriminately adding more reference points is not beneficial.

We also provide the full results of ablations on the designs in LaneSeg decoder in Tab. 8. It can be observed that with the inclusion of heads-to-regions and identical initialization mechanisms, the model's performance improves in all aspects. Fig. 6 presents the loss with identical initialization and without identical initialization. It can be speculated that the better performance with the inclusion of identical initialization is possibly due to its demonstrated largely improved loss convergence.

### B.2.3 Impact of Mask Supervision

Regarding the ablation experiment on mask supervision, the mask supervision improves the performance ($+2.0$ $AP_{ls}$ and $+0.6$ $TOP_{lsls}$). This is likely because incorporating instance mask prediction enhances feature representation within the lane segment region. Additionally, the variations in hyperparameter $\lambda_{seg}$ do not have a significant impact on the results.

## C Qualitative Visualization

As shown in Fig. 7, we select three different road structures to showcase the algorithm's performance as the visualization results. The first structure includes a typical intersection, lane segment diverge, and merge. The second structure represents a long-tail scenario of a road around the park with complex ingress and egress. The third structure depicts two T-junctions. LaneSegNet demonstrates superior capability in capturing the geometric variations of these situations compared to other approaches.

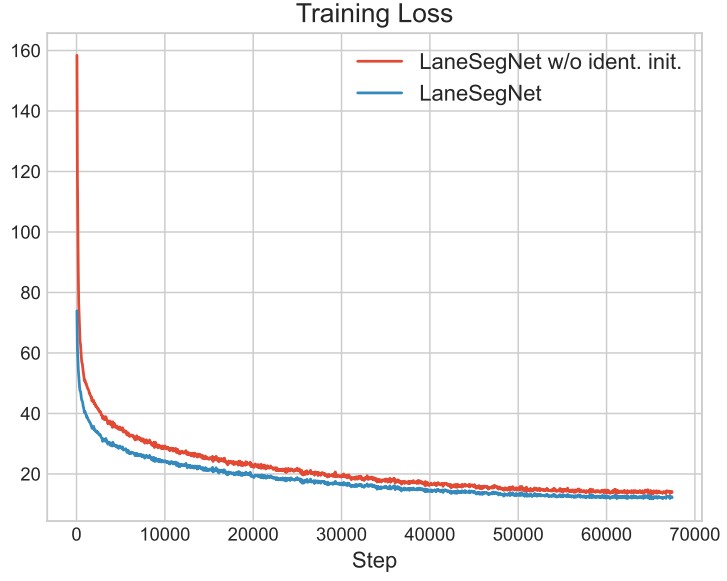

Figure 6: **The loss curves with and without identical initialization**.

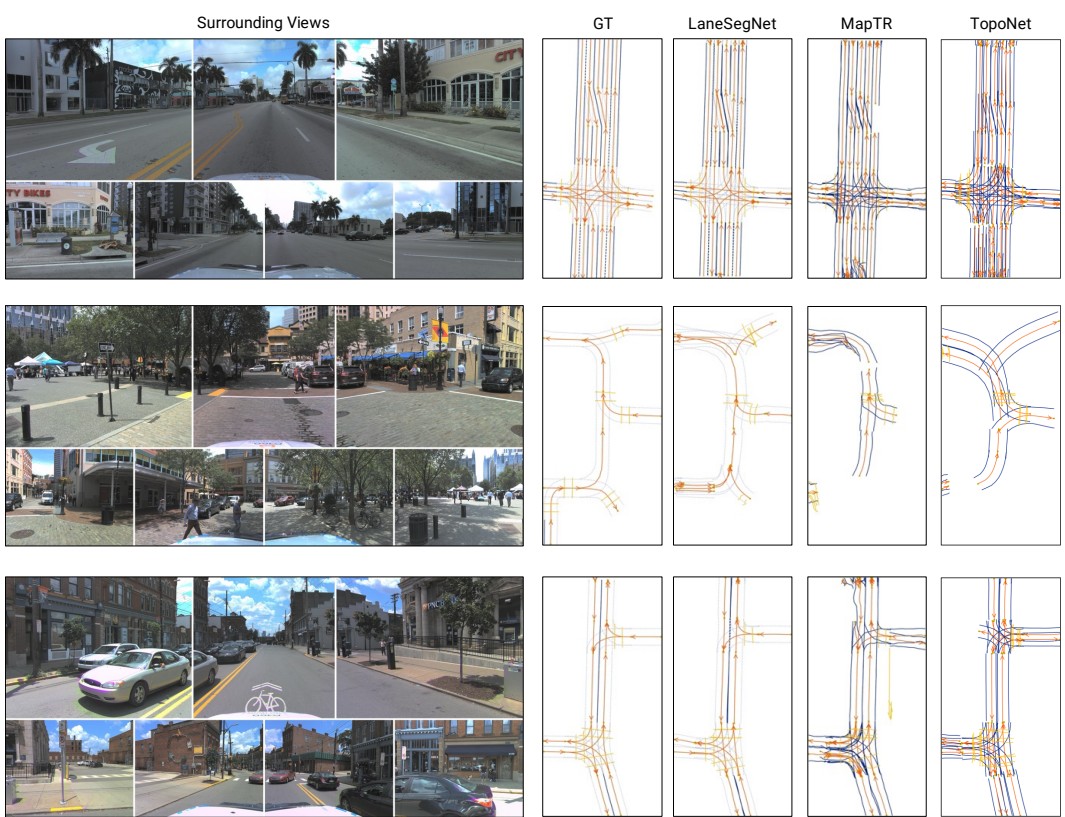

Figure 7: **Qualitative results under different road structures**.

