# OpenReview forum: "LaneSegNet: Map Learning with Lane Segment Perception for Autonomous Driving"
_ICLR.cc/2024/Conference — ICLR 2024 poster_

### Official Review · Reviewer_afCW · 2023-10-23

**Soundness:** 3 good
**Presentation:** 3 good
**Contribution:** 2 fair
**Rating:** 6
**Confidence:** 4

**Summary:**

This paper presents a novel approach to map learning for autonomous driving systems by utilizing the commonly used PV2BEV feature transformation. In contrast to existing methods, this approach introduces a new representation called "lane segment," which incorporates both geometry and topology information. The proposed model is built upon the BEVFormer architecture and incorporates two key modifications: a lane attention module and an identical initialization strategy for reference points, aimed at enhancing the model's prediction capabilities. The prediction branches of the model consist of multiple MLPs, which collectively generate the final predicted lane segment, including the centerline, laneline, laneline type, and adjacent matrix for the lane topology. The effectiveness of the proposed method has been demonstrated through validation on the OpenLaneV2 dataset, showcasing a significant improvement over other existing approaches.

**Strengths:**

1. This paper introduces an innovative end-to-end approach to jointly predict the centerline and laneline scheme, which is a unique contribution compared to existing methods.
2. In contrast to MapTR, which employs hierarchical queries for map element prediction, this method utilizes a single query for both centerline and laneline prediction. The authors propose a heads-to-regions mechanism and distribute reference points evenly within a lane segment, thereby enhancing feature aggregation for both long-range and local image features.
3. By employing an identical initialization strategy, the model achieves remarkable performance on the OpenLaneV2 dataset, demonstrating its effectiveness and high accuracy.

**Weaknesses:**

1. The definition of "long range" is not clearly defined in the paper. The authors mention that the OpenLaneV2 dataset is reannotated using the proposed lane segment manner, resulting in lanes being broken into multiple segments. By using shorter lane segments, I don't think "long range" is challange.
2. Additionally, manually dividing a lane into segments without any visual cues may introduce unnecessary challenges for the model's prediction, whichi is also not critical for autonomous driving systems.
3. In the context of map learning, the novelty of topology prediction in this paper is limited.

**Questions:**

1. MapTRV2 is a stronger baseline but the relative results are not compared in the experiment section.
2. How to innitialized multiple reference points uniformly inside a lane segment through positional query is not discussed detaily.

---

> ### Author Response · Authors · 2023-11-18
> **Author Response for Reviewer afCW**
>
> Thanks for your helpful review. We address your concerns below.
>
> ---
>
> > $\color{brown}{Question 1:}$ The definition of "long range" is not clearly defined in the paper. The authors mention that the OpenLaneV2 dataset is reannotated using the proposed lane segment manner, resulting in lanes being broken into multiple segments. By using shorter lane segments, I don't think "long range" is a challenge.
>
> Thanks for your question. We have analyzed the data of lane segments and the average length is 21.44m and the standard deviation is 15.86. Compared with OpenLane-V2 whose average length is 24.11m and the standard deviation is 18.47, the length of lane segments is only a little bit shorter than that of centerlines. So “long range” still remains a great challenge for lane segments.
>
> Besides, it is also demonstrated in the following results where the increased perception range (30m $\to$ 50m) has led to a significant performance drop in all metrics.
>
> | Setting                                  | mAP  | AP$_{ls}$ | AP$_{ped}$ | TOP$_{lsls}$ | AE$_{type}$ | AE$_{dist}$ |
> | ---------------------------------------- | :--: | :-------: | :--------: | :----------: | :---------: | :---------: |
> | $\pm30\textit{m} \times \pm15\textit{m}$ | 43.3 |   42.5    |    44.2    |     17.7     |     6.7     |    0.536    |
> | $\pm50\textit{m} \times \pm25\textit{m}$ | 32.6 |   32.3    |    32.9    |     8.1      |     9.2     |    0.673    |
>
> ---
>
> > $\color{brown}{Question 2:}$ Additionally, manually dividing a lane into segments without any visual cues may introduce unnecessary challenges for the model's prediction, which is also not critical for autonomous driving systems.
>
> We really appreciate your insight. Actually, our splitting method is not random splitting without any visual cues. We have detailedly introduced the data processing method in Appendix A.2. To be brief, the visual cues can be the divergence, merge, intersection, and changes in the laneline type of two consecutive lanes. Regarding line types, while the data processing currently focuses on the most fundamental line types, such as dashed, solid, and non-visible, it is also flexible enough to support adding other line types if required.
>
> ---
>
> > $\color{brown}{Question 3:}$ In the context of map learning, the novelty of topology prediction in this paper is limited.
>
> Agreed. We haven’t proposed any specific topology module or design to elevate our model’s performance on topology prediction. But thanks to the designing merits of our representation, we have also witnessed a significant performance enhancement in topology prediction as shown in Tab. 3, demonstrating the training friendliness of lane segments. Indeed, we're looking forward to more methods adopting our formulation and exploring the possibilities of employing more novel topological designs in this task.
>
> ---
>
> > $\color{brown}{Question 4:}$ MapTRV2 is a stronger baseline but the relative results are not compared in the experiment section.
>
> Thank you for your feedback. We acknowledge the importance of MapTRv2 as a baseline and have accordingly conducted a comparative analysis, the results of which are presented in the accompanying table. Utilizing the same R50 backbone and BEVFormer encoder as MapTR and LaneSegNet, our findings indicate that while MapTRv2 does demonstrate enhanced performance over MapTR, LaneSegNet continues to outperform MapTRv2 in a fair comparison.  We have added the results into the revised paper.
>
> | Setting    | mAP  | AP$_{ls}$ | AP$_{ped}$ | AE$_{dist}$ |
> | ---------- | :--: | :-------: | :--------: | :---------: |
> | MapTR      | 27.0 |   25.9    |    28.1    |    0.695    |
> | MapTRv2    | 28.5 |   26.6    |    30.4    |    0.702    |
> | LaneSegNet | 32.6 |   32.3    |    32.9    |    0.673    |
>
> ---
>
> > $\color{brown}{Question 5:}$ How to innitialize multiple reference points uniformly inside a lane segment through positional query is not discussed detailedly.
>
> Thank you for your insightful comment. In response to your suggestion, we have enhanced the details in the revised manuscript
>
> In the process of heads-to-regions, it is critical to guarantee that reference points are uniformly dispersed within a lane segment. To achieve this, we methodically allocate four reference points along two predicted boundaries of a lane segment. Consequently, lane attention is directed to each local area along the lane segment through distinct heads, enabling long range attention and precise local feature extraction.
>
> In the first decoder layer, due to the absence of geometric guidance from previous layers' predictions, the identity initialization mechanism generates only one reference point from each positional query embedding. Thus, the eight reference points utilized in the first layer's lane attention are positioned identically. This deliberate design choice aims to stabilize the learning of positional priors by filtering out the distraction of intricate geometries.

---

### Official Review · Reviewer_jPp9 · 2023-10-31

**Soundness:** 3 good
**Presentation:** 1 poor
**Contribution:** 2 fair
**Rating:** 5
**Confidence:** 4

**Summary:**

This work introduces LaneSegNet for predicting lane segments from multi-view camera images.
They propose a new representation for lane segments and develop two new techniques for map-prediction architectures.
Their lane attention module is an alternative to deformable cross-attention that aims to better capture long-range interactions, and their reference point initialization is a way to better capture spatial priors of map elements.
They perform experiments on the OpenLane-V2 dataset and show improvements on map detection, centerline prediction, and lane segment segmentation.

**Strengths:**

* Strong results on the new OpenLane-V2 dataset. Numbers are solid and their tiny model outperforms  the other baselines.
* I like the general trend of work that directly predicted structured representations for mapping.

**Weaknesses:**

* Motivation for a new task - I'm not convinced this task is necessary. Can the authors elaborate why this is a more suitable representation as opposed to the representations in VectorMapNet, MapTR, others?
* Writing - the majority of Section 3 (method) was not detailed enough. Section 3.3 (training loss) in particular describes several losses very briefly and with no citations.
* Comparison with prior work - OpenLane-V2 is relatively new and baseline numbers are from authors implementations. It would help calibrate numbers if they applied their architecture to the original tasks conducted in nuScenes.
* Figures should be more informative - Figure 2 looks like a generic map prediction architecture and Figure 3 does not help illustrate the lane attention module.

**Questions:**

* Take for example a simple road with two lanes (V1, V2 from left to right) - with the proposed representation, this would be represented as V1 = {V1_left, V1_center, V1_right} and V2 = {V2_left, V2_center, V2_right}.
How is the consistency of V1_right and V2_left enforced?
It seems redudant to regress additional left/right lanes instead of simply regressing centerline + adding left/right lane id attributes.

* Why is Argoverse 2 mentioned on page 7?

* Metrics: it would be helpful to readers to define the acronyms for OLS and DET

---

> ### Author Response · Authors · 2023-11-18
> **Author Response for Reviewer jPp9 (1/2)**
>
> Thanks for your insightful suggestions and valuable advice and we really appreciate your comments. We address your questions below.
>
> ---
>
> > $\color{brown}{Question 1:}$ Motivation for a new task - I'm not convinced this task is necessary. Can the authors elaborate why this is a more suitable representation as opposed to the representations in VectorMapNet, MapTR, others?
>
> We agree that the map-element-wise abstraction adopted by VectorMapNet and MapTR provides a suitable preliminary representation for online mapping. However, as shown in Fig.1(b),  the representation of map elements fails to adequately indicate topology relationships at intersections and/or situations where lanes diverge/merge. But as demonstrated in previous works such as STSU [1], OpenLane-V2 [2], TopoMLP [3], SMERF [4], etc., such topology relationships are very crucial for autonomous driving, without which vehicles cannot trustfully predict its next action such as changing lanes legally (in this case), turning around and so on. Indeed, lane segment can greatly combine the advantages of both representations.
>
> > [1] Can Y B, Liniger A, Paudel D P and Van Gool L, Structured bird’s-eye-view traffic scene understanding from onboard images. In ICCV, 2021.
> > [2] Wang H, Li T, Li Y, et al. Openlane-v2: A topology reasoning benchmark for scene understanding in autonomous driving. In NeurIPS Track Datasets and Benchmarks, 2023.
> > [3] Wu D, Chang J, Jia F, et al. TopoMLP: An Simple yet Strong Pipeline for Driving Topology Reasoning. arXiv preprint arXiv:2310.06753, 2023.
> > [4] Luo K Z, Weng X, Wang Y, et al. Augmenting Lane Perception and Topology Understanding with Standard Definition Navigation Maps. arXiv preprint arXiv:2311.04079, 2023.
>
> ---
>
> > $\color{brown}{Question 2:}$ Writing - the majority of Section 3 (method) was not detailed enough. Section 3.3 (training loss) in particular describes several losses very briefly and with no citations.
>
> Thanks for your valuable advice. Considering that the encoder, predictor, and losses applied in our framework are universally adopted by previous works, we saved such space for those more important things in our first submission. More implementation information has already been added to Appendix A.3, including a more detailed explanation of our training loss. We've also accordingly added the following related citations and corresponding references to Sec.3.3 Training Loss in the manuscript:
>
> > [1] Milletari F, Navab N, Ahmadi S A. V-net: Fully convolutional neural networks for volumetric medical image segmentation. In 3DV. 2016.
> > [2] Lin T Y, Goyal P, Girshick R, et al. Focal loss for dense object detection. In ICCV. 2017.
>
> ---
>
> > $\color{brown}{Question 3:}$ Comparison with prior work - OpenLane-V2 is relatively new and baseline numbers are from authors implementations. It would help calibrate numbers if they applied their architecture to the original tasks conducted in nuScenes.
>
> Thanks and Agreed. We have re-implemented related experiments on nuScenes' original individual task--centerline detection and the results are shown in the following table. Compared with our baseline TopoNet, LaneSegNet outperforms by +6.1 in DET$\_{l}$, +6.4 in TOP$\_{ll}$, and +10.0 in OLS, which is consistent with our experiment results in OpenLane-V2.
>
> | Method         | DET$_{l}$ | TOP$_{ll}$ |   OLS    |
> | -------------- | :-------: | :--------: | :------: |
> | MapTR          |   8.3     |    0.1     |   11.5   |
> | TopoNet        |   24.3    |    2.5     |   20.1   |
> | **LaneSegNet** | **30.4**  |  **8.9**   | **30.1** |
>
> ---
>
> > $\color{brown}{Question 4:}$ Figures should be more informative - Figure 2 looks like a generic map prediction architecture and Figure 3 does not help illustrate the lane attention module.
>
> Thanks for your valuable suggestion. To fairly validate the representation advantages, we adopt a generic map prediction architecture as shown in Figure 2. We intend to modify Figure 2 to to better highlight the unique aspects of our design in the revised paper. Additionally, we are enthusiastic about the potential for future methods to adopt and expand upon our representational ideology, leading to innovative designs.
>
> Regarding Figure 3, we have made significant revisions to provide a clearer depiction of the lane attention module, particularly emphasizing the heads-to-region mechanism. Detailed implementation aspects and further illustrations have also been incorporated into Appendix A.1.3 to enhance understanding. We appreciate your valuable contribution to improving our paper.

---

> ### Author Response · Authors · 2023-11-18
> **Author Response for Reviewer jPp9 (2/2)**
>
> > $\color{brown}{Question 5:}$ Take for example a simple road with two lanes (V1, V2 from left to right) - with the proposed representation, this would be represented as V1 = {V1_left, V1_center, V1_right} and V2 = {V2_left, V2_center, V2_right}. How is the consistency of V1_right and V2_left enforced? It seems redundant to regress additional left/right lanes instead of simply regressing centerline + adding left/right lane id attributes.
>
> Thank you for highlighting this question. Indeed, there can be instances of inconsistency between V1$\_{right}$ and V2$\_{left}$ in our proposed representation. To address this, we have incorporated advanced post-processing techniques, such as the union-find algorithm, which have demonstrated efficacy in resolving these discrepancies. Actually, such a problem may also exists in other online mapping algorithms, whether it is MapTR, PivotNet, or any other previous methods. Direct results of such map learning methods are usually redundant under a fixed confidence threshold. They may also require corresponding post-processing to simplify the output.
>
> With respect to the second question, it's true that regressing centerline + adding left/right id attributes can integrate enough semantic information, but the merits of performance enhancement brought by the additional geometrical supervision will be deprived at the same time. In addition, we have also conducted such experiments but insight comes that a strong reasoning capability is strongly bound with a strong prediction performance and thus the absence of supervision of the left/right lane divider leads to a drop in the model's reasoning performance in previous experiments as shown in Tab. 4.
>
> ---
>
> > $\color{brown}{Question 6:}$ Why is Argoverse 2 mentioned on page 7?
>
> The OpenLane-V2 dataset is partially built upon the Argoverse 2 dataset. So the original map data has to be firstly re-collected from the Argoverse 2 dataset before further alignment or processment. More details of the data processing approach are elaborated in Appendix A.2.
>
> ---
>
> > $\color{brown}{Question 7:}$ Metrics: it would be helpful to readers to define the acronyms for OLS and DET.
>
> Thanks for your advice. OLS is short for *OpenLane-V2 Score* which is a comprehensive metric defined in OpenLane-V2 while DET represents the detection metric of certain attributes which are usually given in subscripts and calculated in the way of mAP. We've added this paraphrase to the manuscript in the form of footnotes on page 7.

---

### Official Review · Reviewer_eWbz · 2023-11-01

**Soundness:** 2 fair
**Presentation:** 3 good
**Contribution:** 2 fair
**Rating:** 5
**Confidence:** 3

**Summary:**

This paper proposes LaneSegNet to annote lane segment for online end-to-end map learning. The network leverages both geometry and topology. It introduces a lane detection module with heads-to-regions for long range attention and identical initialization of reference points to stablize the training.

**Strengths:**

1. The results look strong.
2. Abalation studies were conducted to compare different choices of attentions and initialization.

**Weaknesses:**

1. The model uses one-to-one optimal assignment between predictions and ground truth with the Hungarian algorithm, which is usually slow and unstable and difficult to train, particularly in the beginning.
2. While metrics look promising, the metrics are still pretty low. It would be more interested in showing what type of lanes the model can do  so well that it can be used for autonomous driving.

**Questions:**

1. How long is each lane segment ?
2. For a lane segment {A_left, A, A_right}, each appears in 3 lane segments, how to combine them to get final result for e.g. A ?

---

> ### Author Response · Authors · 2023-11-18
> **Author Response for Reviewer eWbz**
>
> Thanks for your valuable comments. We address your concerns below.
>
> ---
>
> > $\color{brown}{Question 1:}$ The model uses **one-to-one optimal assignment** between predictions and ground truth with the Hungarian algorithm, which is usually slow and unstable and difficult to train, particularly in the beginning.
>
> Thanks for your suggestion. In our experiments, we adopted the one-to-one optimal assignment strategy from the DETR [1] model, which helps eliminate the need for post-processing like non-maximum suppression (NMS). However, as you noted, this strategy is usually slow and unstable and difficult to train, particularly in the beginning. To address this, we've refined the Deformable Attention [2] module with lane attention and identical initialization. These adjustments have improved the model's convergence, as shown in our results Tab. 8. Additionally, we are considering the one-to-many assignment strategy as a future exploration direction.
>
> > [1] Carion N, Massa F, Synnaeve G, et al. End-to-end object detection with transformers. In ECCV, 2020.
> > [2] Zhu X, Su W, Lu L, et al. Deformable detr:
> Deformable transformers for end-to-end object detection. In ICLR, 2020.
>
> ---
>
> > $\color{brown}{Question 2:}$ While metrics look promising, the metrics are still pretty low. It would be more interested in showing what type of lanes the model can do so well that it can be used for autonomous driving.
>
> The low metrics are primarily due to two factors. First, compared to existing HD map learning tasks, whose perception range is $\pm30\textit{m} \times \pm15\textit{m}$ , we have expanded the range to $\pm50\textit{m} \times \pm25\textit{m}$. Second, the metrics are defined relatively stringently and compactly.
>
> Taking two main metrics AP$\_{ls}$ and TOP$\_{lsls}$ as examples:
>
> - The calculation of AP$\_{ls}$ is based on the lane segment distance between the predicted point sets and the ground truth in Equation 5. As we have expanded the range, the distance differences in point sets farther away make a great difference by leading to an excess of the matching threshold and an increase in false positives thus resulting in lower metrics.
>
> - The TOP$\_{lsls}$ metric requires both accurate line detection and the detection of relevant relationships, making its assessment criteria quite stringent and compact. It is indicated by previous large amounts of experiments that the upper bound of TOP$\_{lsls}$ is about the numerical level of AP$\_{ls}$‘s square.
>
> To further illustrate this issue, we directly re-evaluated our trained model under the setting of $\pm30\textit{m} \times \pm15\textit{m}$, and the results are as follows.
>
> | Setting                                  | mAP  | AP$_{ls}$ | AP$_{ped}$ | TOP$_{lsls}$ | AE$_{type}$ | AE$_{dist}$ |
> | ---------------------------------------- | :--: | :-------: | :--------: | :----------: | :---------: | :---------: |
> | $\pm30\textit{m} \times \pm15\textit{m}$ | 43.3 |   42.5    |    44.2    |     17.7     |     6.7     |    0.536    |
> | $\pm50\textit{m} \times \pm25\textit{m}$ | 32.6 |   32.3    |    32.9    |     8.1      |     9.2     |    0.673    |
>
> In addition, we provide qualitative visualization in Appendix C. We can see that laneSegNet performs well as shown in Figure 7 no matter whether it is in typical, complex, or long-tail scenarios.
>
> ---
>
> > $\color{brown}{Question 3:}$ How long is each lane segment?
>
> Thanks for your insightful question. We have analyzed the data of lane segments. The average length is 21.44m and the standard deviation is 15.86. Compared with OpenLane-V2 whose average length is 24.11m and the standard deviation is 18.47, the continuity of lanes is still well-preserved.
>
> ---
>
> > $\color{brown}{Question 4:}$ For a lane segment {A_left, A, A_right}, each appears in 3 lane segments, how to combine them to get final result for e.g. A?
>
> According to Sec.3.1, the geometric composition of a lane segment is defined as the vectorized centerline and its corresponding lane boundaries, while its semantics include lane segment classification and line type of the left/right lane boundary. During map learning, the decoder outputs an instance embedding corresponding to each lane segment, followed by predictors that forecast the centerline and boundaries' offset relative to the centerline, along with its associated semantics. Consequently, for the learning of a lane segment, these pieces of information are acquired through instance embedding and are inherently interlinked.

---

### Official Review · Reviewer_VcjB · 2023-11-02

**Soundness:** 3 good
**Presentation:** 4 excellent
**Contribution:** 3 good
**Rating:** 8
**Confidence:** 4

**Summary:**

The work presents a lane-segment based approach for map constructions. The goal is to generate a comprehensive representation [ Map elements and centerline combined] of the lanes within the map. The results are competitive with previous approaches with a single model generalizing to multiple lane based tasks. The method achieves non-marginal results on different tasks.

**Strengths:**

- The paper is well-written. Detailed explanation of each component. Starting from the introduction to a comprehensive related work review. Then section 3.2 details the approach.
- Experiments are comprehensive, and non marginal improvments in the evaluation metrics can be seen
- Ablation study for the different choices of attention mechanism and other design choices validating the model architecture is available.
- The regression in AP_div is discussed in section 4.3

—
I increased my score after taking a look into rebuttal and other reviewers. The work is sound in terms of value and a novelty of the concept. It’s true that the new concept is having a form of disadvantage but the results are promising. The results indeed generalize to other datasets as shown in the rebuttal.

**Weaknesses:**

- It seems there is only one dataset that contains all of the tasks. What about the performance on other datasets where such tasks individually are available.
- No ablation on the choice for the BEV encoder.

**Questions:**

It would be nice to see the impact of such a map on downstream tasks such as motion planning/prediction.

---

> ### Author Response · Authors · 2023-11-18
> **Author Response for Reviewer VcjB**
>
> Thanks for your insightful perspective. We have thoughtfully considered and would like to respond to each of your points as follows.
>
> ---
>
> > $\color{brown}{Question 1:}$ It seems there is only one dataset that contains all of the tasks. What about the performance on other datasets where such tasks individually are available.
>
> Thanks for your observation. Indeed, the Argoverse 2 dataset uniquely meets our requirements for constructing lane segment labels. We agree that the results on datasets where such tasks are individually available would be convincing and we really appreciate your suggestion.
>
> Consequently, we conducted further comparison experiments using the OpenLane-V2 subset-B benchmark, derived from the nuScenes dataset. This benchmark includes only centerline and lane graph annotations. To address this, we produced the pseudo-label of lane segments by assigning a standardized lane width to the lane centerlines. The results demonstrate that LaneSegNet still outperforms other state-of-the-art methods in this scenario.
>
> | Method         | DET$_{l}$ | TOP$_{ll}$ |   OLS    |
> | -------------- | :-------: | :--------: | :------: |
> | MapTR          |   8.3     |    0.1     |   11.5   |
> | TopoNet        |   24.3    |    2.5     |   20.1   |
> | **LaneSegNet** | **30.4**  |  **8.9**   | **30.1** |
>
> > The results of TopoNet are sourced directly from the official results published in OpenLane-V2 paper.
> > The OLS metric is only calculated on centerline perception aspect.
>
> ---
>
> > $\color{brown}{Question 2:}$ No ablation on the choice for the BEV encoder.
>
> Thanks. We have added an ablation study on the BEV encoder section. For implement convenience, we selected GKT encoder to compare with BEVFormer encoder.
>
> | Method               |   mAP    | AP$_{ls}$ | AP$_{ped}$ | TOP$_{lsls}$ | AE$_{type}$ | AE$_{dist}$ |
> | :------------------- | :------: | :-------: | :--------: | :----------: | :---------: | :---------: |
> | GKT [1]              |   30.1   |   28.9    |    30.3    |     6.6      |    10.9     |    0.700    |
> | **BEVFormer (Ours)** | **32.6** | **32.3**  |  **32.9**  |   **8.1**    |   **9.2**   |  **0.673**  |
>
> > [1] Chen S, Cheng T, Wang X, et al. Efficient and robust 2d-to-bev representation learning via geometry-guided kernel transformer. arXiv preprint arXiv:2206.04584.
>
> ---
>
> > $\color{brown}{Question 3:}$ It would be nice to see the impact of such a map on downstream tasks such as motion planning/prediction.
>
> Agreed. In the context of downstream prediction and planning tasks in autonomous driving, the standard of map input formats for map encoding modules remains a topic of debate. While LaneGCN [1] utilizes a lane graph for motion forecasting, VectorMap [2] employs road elements as inputs. Consequently, we propose that the integration of both features in a lane segment representation would better cater to the diverse needs of prediction algorithms. Regarding the planning module, we consider the widely-used Lattice Planner [3] as an example. Its loss function design includes constraints for both lane boundary adherence and following centerline, demonstrating the necessity of a comprehensive and unified expression in lane segments. Additionally, the integration of LaneSegNet within an end-to-end planning paradigm such as UniAD [4] is an intriguing prospect that we plan to explore in future work.
>
> > [1] Liang M, Yang B, Hu R, et al. Learning lane graph representations for motion forecasting. In ECCV, 2020.
> > [2] Gao J, Sun C, Zhao H, et al. Vectornet: Encoding hd maps and agent dynamics from vectorized representation. In CVPR, 2020.
> > [3] Werling M, Ziegler J, Kammel S, et al. Optimal trajectory generation for dynamic street scenarios in a frenet frame. In ICRA, 2010.
> > [4] Hu Y, Yang J, Chen L, et al. Planning-oriented autonomous driving. In CVPR, 2023.

---

> > ### Comment · Reviewer_VcjB · 2023-11-18
> >
> > Thanks, I went through your answers and combined other reviewers rebuttal. I’m increasing my score accordingly.

---

### Author Response · Authors · 2023-11-18
**General Response to Area Chairs and Reviewers**

Dear Area Chairs and Reviewers,

We appreciate all the reviewers for their careful reviews and valuable comments. We have taken each comment into consideration and have made revisions to our paper accordingly. The revised texts are marked in blue. Our responses to specific concerns are detailed below, and we have focused on adding clarity and depth to address any ambiguities. We are grateful for the opportunity to improve our work with your guidance.

Thanks.

---

> ### Author Response · Authors · 2023-11-22
> **Invitation for Further Discussion**
>
> Dear Reviewers,
>
> We would like to express our sincere gratitude again for your valuable comments and thoughtful suggestions. We also thank the reviewers for acknowledging our strengths and contributions, such as the novel and interesting design of the concept (VcjB, afCW), important research topic on direct structured map learning (jPp9), innovative design of the model architecture and attention mechanism (VcjB, afCW), promising results for each module and solid ablations (VcjB, eWbz, jPp9, afCW), and well-written (VcjB).
>
> Throughout the rebuttal phase, we tried our best to address concerns, augment experiments to fortify the paper, and refine details in alignment with your constructive feedback (See changes marked as blue in the revised version). Since the discussion time window is very tight and is approaching its end, we truly hope that our responses have met your expectations and addressed any concerns. We genuinely do not want to miss the opportunity to engage in further discussions with you, which we hope could contribute to a more comprehensive evaluation of our work. Should any lingering questions persist, we are more than willing to offer any necessary clarifications.
>
> With heartfelt gratitude and warmest regards,
>
> The Authors

---

### Meta-Review · Area_Chair_fhD5 · 2023-12-15

**Metareview:**

This paper introduces a lane-segment-based approach for map constructions, aiming to generate a comprehensive representation of lanes within a map. The method achieves non-marginal results on different tasks and leverages both geometry and topology. The paper introduces LaneSegNet for online end-to-end map learning, introducing a lane detection module and identical initialization of reference points. The researchers also introduce a new representation called "lane segment" for autonomous driving systems, incorporating both geometry and topology information. The model is built upon the BEVFormer architecture and includes multiple MLPs to generate the final predicted lane segment.

## Strengths

• The paper provides detailed explanations of each component, from introduction to related work review.
• Experiments show non-significant improvements in evaluation metrics.
• Ablation studies validate model architecture choices.
• Regression in AP_div is discussed in section 4.3.
• Results show strong performance on the OpenLane-V2 dataset.
• The paper introduces an innovative end-to-end approach for jointly predicting centerline and laneline schemes.
• The method uses a heads-to-regions mechanism and distributes reference points evenly within a lane segment.
• The model achieves remarkable performance on the OpenLaneV2 dataset.

## Weaknesses

• The model uses a single dataset for tasks, limiting performance on other datasets.
• The choice for the BEV encoder is not clear.
• The model uses a slow, unstable Hungarian algorithm for optimal assignment between predictions and ground truth.
• The metrics are promising but low, indicating the model's potential for autonomous driving.
• The authors' motivation for a new task is questionable.
• The writing is not detailed enough, with sections describing losses briefly and without citations.
• Comparison with prior work is suggested, especially considering OpenLane-V2's relatively new nature.
• The figures should be more informative, with Figure 2 and 3 lacking in lane attention module illustrations.
• The definition of "long range" is unclear, and manual lane segmentation without visual cues may introduce unnecessary challenges.
• The novelty of topology prediction in map learning is limited.

**Justification For Why Not Higher Score:**

The paper shows significant improvement to be accepted thanks to all the addressed concerns.

**Justification For Why Not Lower Score:**

All reviewers agreed that the paper is good enough to be accepted.

---

### Decision · Program_Chairs · 2024-01-16

Accept (poster)